# Blind Deconvolution Based on Compressed Sensing with bi-*l*_0_-*l*_2_-norm Regularization in Light Microscopy Image

**DOI:** 10.3390/ijerph18041789

**Published:** 2021-02-12

**Authors:** Kyuseok Kim, Ji-Youn Kim

**Affiliations:** 1Department of Radiation Convergence Engineering, Yonsei University, Gangwon-do 26493, Korea; seokkyu502@gmail.com; 2Department of Dental Hygiene, College of Health Science, Gachon University, Incheon 21936, Korea

**Keywords:** blind deconvolution, bi-*l*_0_-*l*_2_-norm regularization, compressed sensing, qualitative and quantitative analyses, light microscopy image

## Abstract

Blind deconvolution of light microscopy images could improve the ability of distinguishing cell-level substances. In this study, we investigated the blind deconvolution framework for a light microscope image, which combines the benefits of bi-*l*_0_-*l*_2_-norm regularization with compressed sensing and conjugated gradient algorithms. Several existing regularization approaches were limited by staircase artifacts (or cartooned artifacts) and noise amplification. Thus, we implemented our strategy to overcome these problems using the bi-*l*_0_-*l*_2_-norm regularization proposed. It was investigated through simulations and experiments using optical microscopy images including the background noise. The sharpness was improved through the successful image restoration while minimizing the noise amplification. In addition, quantitative factors of the restored images, including the intensity profile, root-mean-square error (RMSE), edge preservation index (EPI), structural similarity index measure (SSIM), and normalized noise power spectrum, were improved compared to those of existing or comparative images. In particular, the results of using the proposed method showed RMSE, EPI, and SSIM values of approximately 0.12, 0.81, and 0.88 when compared with the reference. In addition, RMSE, EPI, and SSIM values in the restored image were proven to be improved by about 5.97, 1.26, and 1.61 times compared with the degraded image. Consequently, the proposed method is expected to be effective for image restoration and to reduce the cost of a high-performance light microscope.

## 1. Introduction

Light microscopy based on magnification using an objective lens is widely used as one of the simplest and easiest methods for observing various tissues at the cellular level. Among the techniques for image acquisition, the use of a microscope is a direct method useful for observing and analyzing morphological changes with respect to various conditions. Although a light microscope can be used to most easily observe the changes in the tissue, it is limited in observing fine parts owing to the limitations of spatial resolution and noise amplification. Thus, it is essential to improve the qualities of light microscopic images for accurate analysis [1]. The approach of development of hardware and software technologies is mainly used to improve the microscopic image quality. The development field of hardware technology for improving microscopic images is the most suitable method to achieve a desired level of image quality, but has a large disadvantage of high cost. Thus, methods using image processing technologies are actively studied [2,3,4].

Generally, a light microscopy image can be mathematically modelled as a linear-translation-invariant system incorporating the background noise [5]:(1)$$g\left(x,y\right)=psf\left(x,y\right)⊗⊗f\left(x,y\right)+N\left(x,y\right)$$
where *g* is the image obtained by containing distortion components in the *x*- and *y*-coordinates, *f* is a clean image, *psf* is the point-spread function (PSF), which indicates the degree of blurring occurring owing to the finite imaging system (e.g., focal spot, pixel size), and ⊗⊗ represents the two-dimensional (2D) convolution operator, which is used to obtain the $f\left(x,y\right)$ by predicting and measuring the $psf\left(x,y\right)$ to perform the deconvolution. *N* is the background signal dominated by Poisson and Gaussian noises. In light microscopy images, obtained by numerous photons, the Poisson noise can be assumed as an additive Gaussian noise [6]. Therefore, it is very important to perform an accurate *psf* prediction while effectively controlling the noise.

To overcome these difficulties, image deconvolution algorithms have been studied to obtain successfully restored images under the conditions of an unknown PSF containing the image degradation rate of the imaging system, including the Bayesian method, inverse Radon transform, iteration methods, and deep-learning approaches [7,8,9,10,11,12,13]. These methods were used to obtain a considerably effective result for the ill-posed problem. However, one of the main difficulties of these methods is associated with the presence of noise in the microscopy image, because the noise component prompts an imperfect edge detection and often leads to an unstable or false solution owing to finding the inexact PSF. Moreover, the noise should be amplified by the deconvolution process as a side effect.

A typical approach to controlling the noise while implementing the deconvolution method is to solve the inverse problem using appropriate regularization terms,
(2)$$\hat{f}\left(x,y\right)=\underset{f,psf}{\mathrm{argmin}}\|psf\left(x,y\right)⊗⊗f\left(x,y\right)-g{\left(x,y\right)\|}_{2}^{2}+R\left(f\right)$$
where $R\left(f\right)$ is the regularization term used to compensate the noise component to calculate the error in the part of $\|psf\left(x,y\right)⊗⊗f\left(x,y\right)-g{\left(x,y\right)\|}_{2}^{2}$. This approach enables effective PSF estimation and image restoration, even in images with noise. $R\left(f\right)$ is observed in several *l*_p_-norm-based priors including the *l*_p_-norm-based prior (*p* decreases at regular intervals while iterating) [14], normalized sparsity-based image prior [15], approximate *l*_0_-norm-based image prior [16], and reweighted *l*_2_-norm-based image prior [17]. However, these approaches have limitations, including the staircase artifacts (cartooned artifacts) in *l*_2_-norm-based minimization [18] and noise amplification in sparse deconvolution by *l*_1_-norm-based minimization [19]. Shao et al. [20] introduced the bi-*l*_0_-*l*_2_-norm regularization strategy with an effective performance decreasing the computational cost to restore the degraded images. Their strategy improves the accuracy of the kernel estimation owing to the strategy of evasion from the background noise.

The objective of this study was to apply the bi-*l*_0_-*l*_2_-norm regularization strategy in light microscopy. For that objective, quantitative evaluations, including visual assessment, intensity profile, root-mean-square error (RMSE) [21], edge preservation index (EPI) [22], structural similarity index measure (SSIM) [23], and normalized noise power spectrum (NNPS) [24] were used in both simulation and experiment images. We used the compressed sensing (CS) [25,26,27,28] and conjugate gradient (CG) [29] methods to solve the optimization problem with a high accuracy. Section 2 briefly describes the bi-*l*_0_-*l*_2_-norm regularization strategy in blind deconvolution and study conditions. Section 3 presents the results and discussion, while Section 4 summarizes the conclusions of this study.

## 2. Proposed Blind Deconvolution of the Microscopy Image

Figure 1 shows the entire flowchart of the proposed method incorporating the bi-*l*_0_-*l*_2_-norm-based regularization strategy for an accurate blind deconvolution of a light microscopy image.

This method can be divided into two main parts. (1) The PSF is estimated using the CG method and (2) the restored image is predicted using the CS method. In the blind deconvolution scheme, both PSF and restored image are simultaneously and recursively optimized through the two iterative loops. Briefly, the degraded image *g* acquired by the light microscopy imaging system is assumed to be the current image $f_{c}{}^{\left(0\right)}$ before entering the proposed blind deconvolution scheme. In the first step, the latent PSF of the system is estimated. We use the CG-based framework to obtain the PSF between the Laplacian image of $f_{c}{}^{\left(0\right)}$ and *g*, according to Kim et al. [12],
$$\nabla^{2}g_{c}=\nabla^{2}f_{c}{}^{\left(k\right)}⊗⊗psf_{c}{}^{\left(b\right)},$$
(3)$$psf_{c}{}^{\left(b+1\right)}=\underset{psf_{c}{}^{\left(b\right)}\in Q}{\mathrm{argmin}}\frac{1}{2}{\left(psf_{c}{}^{\left(b\right)}\right)}^{T}⊗⊗\nabla^{2}f_{c}{}^{\left(k\right)}⊗⊗psf_{c}{}^{\left(b\right)}-{\left(psf_{c}{}^{\left(b\right)}\right)}^{T}⊗⊗\nabla^{2}g_{c},$$
where $\nabla^{2}$ is the Laplacian operator, and the updated PSF, $psf_{c}{}^{\left(b+1\right)}$, can be effectively predicted in a short time through CG-based approach. Here, *c* is the red–green–blue (RGB) color and $psf_{c}{}^{\left(b+1\right)}$ calculated for each color channel. And then, $f_{c}{}^{\left(k+1\right)}$ is calculated to obtain the objective function using the CS-based framework with the bi-*l*_0_-*l*_2_-norm-based regularization,
$$f_{c}{}^{\left(k+1\right)}=\underset{f_{c}{}^{\left(k\right)}\in Q}{\mathrm{argmin}}\frac{1}{2}\|f_{c}{}^{\left(k\right)}⊗⊗psf_{c}{}^{\left(b+1\right)}\left(x,y\right)-g_{c}{\|}_{2}^{2}+\alpha R\left(f_{c}{}^{\left(k\right)}\right),$$
(4)$$R\left(f_{c}{}^{\left(k\right)}\right)=\beta_{f}\left(\|\nabla f_{c}{{}^{\left(k\right)}\|}_{0}+\frac{\lambda_{f}}{\beta_{f}}\|\nabla f_{c}{{}^{\left(k\right)}\|}_{2}^{2}\right)$$
where *Q* is the feasible set of $f_{c}{}^{\left(k\right)}$, α is the tuning parameter between the fidelity term and regularization term (we used α=500, obtained empirically), ∇ is the first-order linear differential operator, $\beta_{f}$ is positive factors (heuristically, $\beta_{f}=0.25$ used in this study), and $\lambda_{f}$ is fixed to 5. The optimization problem (Equation (4)) can be efficiently solved using the accelerated gradient-projection Barzilai–Borwein (GPBB) method [30]. $f_{c}{}^{\left(k+1\right)}$ is computed to determine the step size with the GPBB strategy for the next updated image. The loop is repeated until the mismatch between $f_{c}{}^{\left(k+1\right)}$ and $f_{c}{}^{\left(k\right)}$ is smaller than the tolerance *ε* (in this work, 20 to 30 iterations required for solving the $\hat{f}$, approximately). When the value is greater than the specified tolerance (in this study, we used 10^−6^, empirically), $psf_{c}{}^{\left(b+1\right)}$ used in Equation (4) is replaced with $psf_{c}{}^{\left(b\right)},$ and the obtained $f_{c}{}^{\left(k+1\right)}$ is replaced with $f_{c}{}^{\left(k\right)}$ in Equation (3).

## 3. Results and Discussion

In the simulation, we use a numerical star phantom (ISO 15775, Siemens star test chart, image size = 2000 × 2000 pixels) to measure the system resolution by the degree of distinction between line pairs. The phantom has a diameter of 400 µm with 36 pairs of spokes (bright and dark regions) with a pixel pitch of 1.7 µm. Figure 2 shows a 3D plot of the PSF used to convolve the clean image. The PSF is based on a Gaussian distribution (kernel size = 51 × 51 pixels, sigma = 1.5 pixels). We perform the convolution to obtain a blurred image using the clean image and PSF. In addition, white noise with a Gaussian distribution (mean = 0, variation = 0.01) is added to the blurred image to synthesize the degraded image.

Microscopic images of haematoxylin–eosin (H–E) staining in a mouse oral mucosa were acquired for this study. The H–E-stained tissue was produced according to the general histological sample preparation, including fixation in 4% paraformaldehyde, paraffin embedding, tissue sectioning (7 µm), and H–E staining. The microscope images were acquired using a Leica series imaging system (Leica Microsystem, Germany), which consisted of a Leica DM500 microscope (40×/0.65 NA/0.31 mm W.D., 100×/1.25 NA/0.10 mm W.D.), Leica ICC50 E camera (Aptina 1/2 inch CMOS sensor, pixel size = 3.2 µm × 3.2 µm, pixel resolution = 2048 × 1536), and Leica LAS EZ software.

The simulations and experiments were performed using MATLAB^TM^ (MathWorks, Natick, MA, USA, R2019a) (computer hardware: central processing unit: Intel, Santa Clara, CA, USA, Xeon Platinum 8168 @ 2.70 GHz; random-access memory: Samsung, South Korea, 8G×4 DDR4 21300; graphics processing unit (GPU): NVIDIA, Santa Clara, CA, USA, GTX 1080 11 GB). The proposed blind deconvolution method required less than 2 s with the GPU parallel processing, which confirms its utility for practical applications.

We measured the profile, RMSE, EPI, SSIM, and NNPS, to evaluate the image quality using the proposed blind deconvolution framework. Among these evaluation parameters, the NPS evaluation parameter was used to calculate the noise of the microscope image based on the distribution in the frequency domain. NPS describes the retention and noise amplitudes of imaging system, including microscopes, and is a parameter that can accurately represent uncertainty and inaccuracies from system signals. In this study, the NPS was measured by introducing the concept of non-uniform gain in a microscopic image shape model with fixed noise. Based on the measured NPS, we normalized to derive the final NNPS result. Regarding the NNPS, the noise characteristic of the image is measured to evaluate the spectrum.
$$NPS\left(u,v\right)=\frac{K^{2}}{N}\langle {\left|ℑ\left\{flat\_area\left(x,y\right)\right\}\right|}^{2}\rangle ,,$$
(5)$$NNPS\left(u,v\right)=\frac{NPS\left(u,v\right)}{{\left(large-area\;signal\right)}^{2}},$$
where K is the detector pixel size, and ℑ represents the Fourier transform operator. The NNPS measures the change in the noise amplitude as a function of the spatial frequency and bridges the noise and spatial resolution characteristics of an image.

Figure 3 shows the simulation results, including the reference image, degraded image (Equation (1)), and image restored using the proposed method. In order to visually analyze the acquired image more clearly, two parts were enlarged and shown below the representative image.

The enlarged images of boxes *B* and *C* indicate that the resolution of the restored image is closer to the resolution of the reference image than to that of the degraded image. By observing the pattern on the edge of the acquired simulation image, we confirmed that blurring occurred in the degraded image in the all areas, and in particular, the restored image showed a very similar pattern to the reference image. In addition, the degree of image degradation on the edge side of the degraded image was confirmed to be such that there was not much difficulty in visual observation. However, as a result of observing center area of the degraded image, it was confirmed that the noise amplification and the patterns were clearly blurred compared to the reference image. As a result of applying the proposed algorithm, the image in the center area was reliably improved, and we can solve the problem of image quality deterioration, which is the most important part when observing a microscopic image of a tissue.

Moreover, Figure 4 shows enlarged patches of the three images in Figure 3 for the background areas marked with box *A*. We confirmed that the patch distribution was obtained differently in each of the three images. In particular, the proposed method was successfully used for blind deconvolution while minimizing the noise amplification.

Figure 5 shows the resultant profile measured along the broken red line in Figure 3 with box *C*. It confirms that the blurring rate of the restored image profile is smaller than that of the degraded image and that the profile of the restored image is similar to that of the reference image. In the profile result graph, it was confirmed that the difference between the reference image and the degraded image was very large at both ends, and the difference tended to decrease as the direction of the pixel position in the middle was increased. Likewise, the profile result of the restored image showed a similar trend as above result, but the width of change was significantly reduced. Moreover, the same tendency is observed in the image evaluation through the quantitative measurement through image quality evaluation factors.

Figure 6 shows the evaluation values obtained using the RMSE, EPI, and SSIM with simulation images.

According to the recently published papers by Saha et al., it can be confirmed that RMSE parameter is used as a quantitative index for the analysis of the utility of deep learning techniques in 3D microscopy images [31]. In addition, according to a study conducted at Linden et al., the RMSE evaluation parameter was used to confirm the difference between true and estimation data when analyzing the usefulness of the image super resolution algorithm for microscope images [32]. In this study, RMSE evaluation parameter was also used to determine how much the proposed method was improved compared to the degraded image when compared with the reference image. The acquired RMSEs are about 0.72 and 0.12 for the degraded and restored images, respectively. In particular, we demonstrated that RMSE value of the image to which the proposed method was applied is approximately six times higher than the value measured in the degraded image.

In microscopic images, when information on an edge area of an acquired tissue image is lost, a situation in which it is difficult to distinguish it from an adjacent tissue often occurs. However, most of the evaluation of the edge information to demonstrate the usefulness of the image processing technology in the microscope image is performed visually or using a profile. Recently, methods for evaluating the sharpness index of an image based on no-reference including the blind/referenceless image spatial quality evaluator or image quality evaluator parameter have been developed and used [33,34,35]. In researches by Kim et al., a no-reference-based method was used to evaluate the noise removal efficiency that is not revealed in the visual evaluation when the denoising algorithm is applied to the light microscopy image [1]. However, since this index has the disadvantage that it cannot accurately measure edge information, it is rarely used as a general method for microscopic image evaluation. Thus, in this study, we tried to evaluate the edge preserving ability of the proposed algorithm using accurate EPI evaluation parameters, which are widely used in other imaging fields. The acquired EPIs are about 0.65 and 0.81 for the degraded and restored images, respectively. In particular, we demonstrated that EPI value of the image to which the proposed method was applied is approximately 1.3 times higher than the value measured in the degraded image. Based on our EPI results, the improvement of the edge information of the tissue light microscopic image is expected to contribute to the precise analysis of the shape of teeth as well as the ability to more clearly determine the boundary between the hard tissue and soft tissue, which are difficult to distinguish.

The SSIM is a method of measuring structural distortion under the assumption that the evaluation of the degree of loss of image quality is caused by the structural distortion of the signal itself rather than by a certain type of error. This method measures image quality through a top-down approach starting from the overall point of view, and can complement the limitations of the error sensitivity approach by observing luminance, contrast, and structure in complex parameters [36]. SSIM can measure the subjectively perceived image quality of distorted images, so it can be widely used in tissue images using a light microscope. The acquired SSIMs are about 0.54 and 0.88 for the degraded and restored images, respectively. We demonstrated that SSIM value of the image to which the proposed method was applied is approximately 1.6 times higher than the value measured in the degraded image. In addition, it was confirmed that the high SSIM value when the proposed algorithm was used in the light microscopic image of the oral mucosa used in this study can achieve more accurate embryology and anatomical information in the local area.

These results indicate that the proposed algorithm well estimates the PSF using the single image while controlling the noise amplification. Figure 7 shows the images of PSF (reference) to make the blurred image and PSF (estimated) using the proposed algorithm. Note that it can be seen that these profiles are almost similar. For the quantitative evaluation, Figure 8 shows that the profiles of two PSF at line AB is almost close.

Here, the profile implemented the Gaussian fitting based on the extracted results. The sigma of PSF (reference) was measured as 1.5 pixels and the that of PSF (estimated) was measured as 1.84 pixels. Accurately estimation of PSF means that successful image restoration is possible and implies that a method of repeatedly updating the PSF and image (blind-deconvolution method) is successfully performed.

Figure 9 shows the experiment results for the degraded image at a magnification of 40, restored image (*l*_1_-norm), and restored image (proposed). To compare the effectiveness of the bi-*l*_0_-*l*_2_-norm-based regularization, we used the *l*_1_-norm-based regularization,
$$f_{c}{}^{\left(k+1\right)}=\underset{f_{c}{}^{\left(k\right)}\in Q}{\mathrm{argmin}}\frac{1}{2}\|f_{c}{}^{\left(k\right)}⊗⊗psf_{c}{}^{\left(b+1\right)}-g_{c}{\|}_{2}^{2}+\alpha R_{1}\left(f_{c}{}^{\left(k\right)}\right),$$
(6)$$R_{1}\left(f_{c}{}^{\left(k\right)}\right)=\beta_{f}\left(\|\nabla f_{c}{{}^{\left(k\right)}\|}_{1}\right)$$

All parameters in Equation (6) are used identically to those in Equation (4). The enlarged image (box *A* in Figure 9) restored using the proposed method exhibits a similar sharpness improvement to that of the comparative image *(**l*_1_-norm-based restored image), at a smaller amplification of the noise. The tendency of these results can be confirmed by images with a higher magnification.

Figure 10 shows examples of the three images at a magnification of 400. We confirmed the improvement of sharpness while suppressing the noise amplification as a result of Figure 9. This result indicates that the proposed algorithm can produce effective results, even in various magnification images. For the quantitative evaluation of noise characteristics, Figure 11 shows that the NNPS characteristics of the restored image (proposed) at spatial frequencies over about 0.10 lp/μm are improved compared to those of the degraded image (×40) and restored image (*l*_1_-norm), owing to the bi-*l*_0_-*l*_2_-norm regularization penalty. These results demonstrate that the proposed method effectively improves the resolution when noise is present in the light microscope image.

In order to compare and evaluate the degree of restoration according to the type of algorithm in the real experimental image, the peak signal to noise ratio (PSNR) and the natural image quality evaluator (NIQE) [34], which is a representative no-reference-based evaluation parameter (Figure 12), were used. Since there is no gold standard image in the real experimental image, a parameter that can compare the whole image was used. We derived PSNR results of 13.10 and 24.18, respectively, when using the conventional *l*_1_-norm method and the proposed method. In addition, the NIQE evaluation results were 11.92 and 4.90, respectively, when the conventional *l*_1_-norm method proposed method were used. In particular, the differences between PSNR and NIQE in the two methods were 1.84 and 2.44 times, respectively. The proposed method, which has small mean square error value that can analyze the restoration rate of an image and the variation between pixels, is expected to be able to better distinguish tissues in microscopic images. In particular, the improved NIQE results are expected to be of great help in applying a deep learning approach that required measuring the distance between features acquired form an image database in the future.

Light microscope is used mainly to generate magnified images of small objects. It has been used in biology because it can be easily operated. However, limitations exist in light microscopy, including the resolution and magnification. A light microscope with transmitted light at a very high magnification can distort the image of a point object [37]. The resolution of the light microscope is a measure of the microscope’s ability to distinguish two adjacent structures. In particular, the size of diffraction and resolving power of a microscopic image are determined by the wavelength and number of apertures of the objective lens. According to these results, it is challenging to clearly distinguish adjacent areas in light microscopy images [38]. One of the approaches to fully utilize the capabilities of the microscope is to obtain a sufficiently high magnification. However, the light microscope has a lower resolution than those of other systems because of the diffuse propagation of the refracted light waves from the lens, which leads to blurred images. Therefore, we applied the bi-*l*_0_-*l*_2_-norm regularization strategy as the image processing technology to improve the qualities of microscopic images. The proposed method will contribute to studies using microscopes as basic systems.

In addition, this study was limited to subjective and quantitative analysis of the applicability of the proposed algorithm in microscopic images. In the future, based on the results of this study, we will conduct a subjective evaluation for users such as imaging experts, doctors, and radiologists to further analyze usefulness.

## 4. Conclusions

This paper presents a blind deconvolution framework for a light microscope image, which combines the benefits of the bi-*l*_0_-*l*_2_-norm regularization with the CS and CG algorithms. It provided a considerable improvement in the resolution while maintaining the noise amplification. According to our simulation and experiment results, the RMSEs of the degraded and restored images were about 0.72 and 0.12, respectively. The EPI of the restored image was about 0.81, about 1.3 times that of the degraded image. The SSIMs for the degraded and restored images were about 0.54 and 0.88, respectively. Moreover, the NNPS characteristics of the restored image (proposed) at spatial frequencies over about 0.10 lp/μm were improved compared to those of the degraded image (×40) and restored image (*l*_1_-norm). These results demonstrate the viability of the proposed method for provision of improved light microscope images. Consequently, the proposed method is expected to be effective for image restoration and can be used to compensate the limitations of imaging hardware in terms of image quality.

## Figures and Tables

**Figure 1 ijerph-18-01789-f001:**
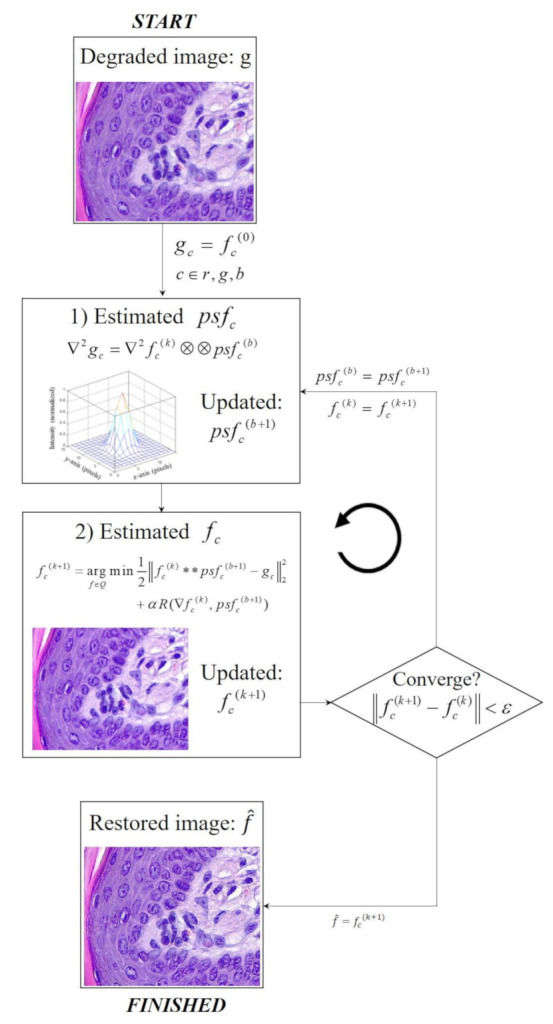
Flowchart of the blind deconvolution using the proposed method with the bi-*l*_0_-*l*_2_-norm regularization of the light microscopy image.

**Figure 2 ijerph-18-01789-f002:**
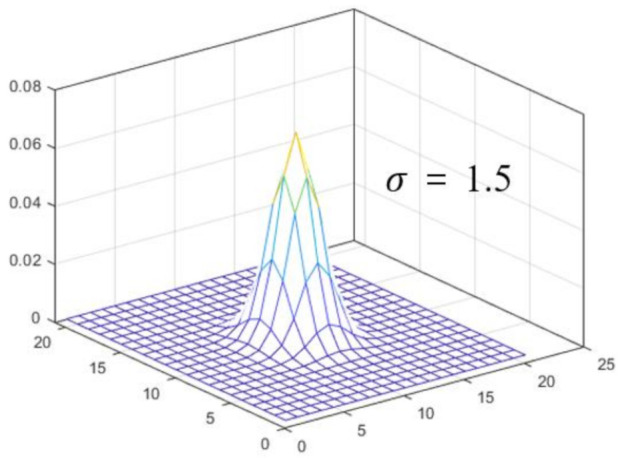
Three-dimensional (3D) plot of the PSF used to convolve the clean image in the simulation study. The PSF is based on a Gaussian distribution with a sigma of 1.5 pixels.

**Figure 3 ijerph-18-01789-f003:**
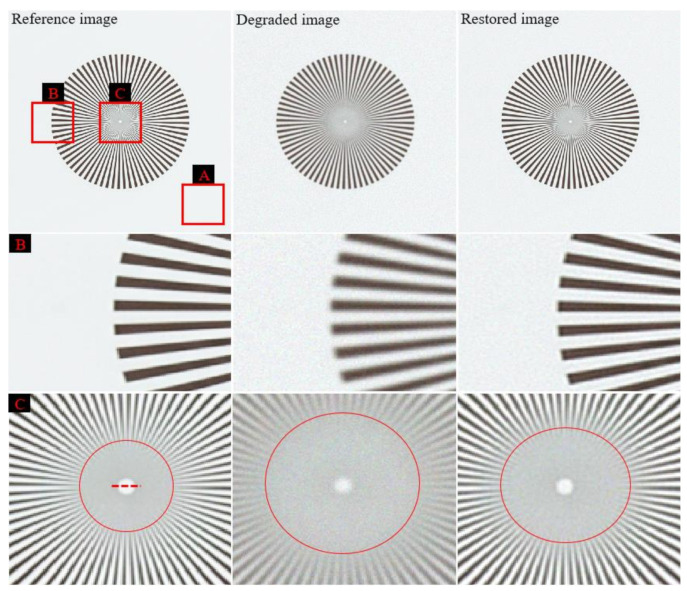
Simulation results including the reference image, degraded image, and image restored using the proposed method. Enlarged images of the regions in boxes *B* and *C* are shown.

**Figure 4 ijerph-18-01789-f004:**
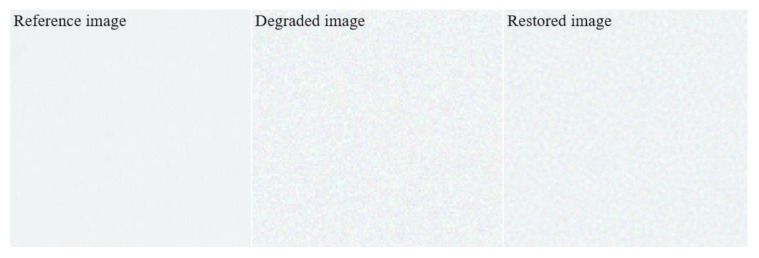
Noise patches of the three images in Figure 3 for the background areas marked with box *A*.

**Figure 5 ijerph-18-01789-f005:**
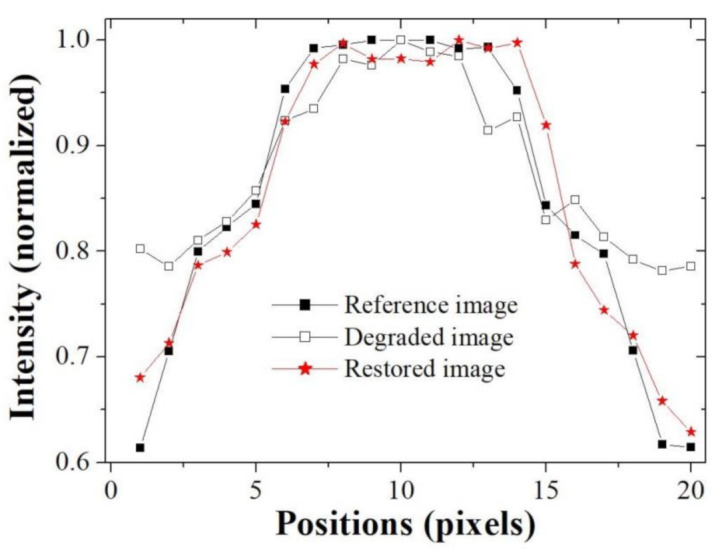
Resultant profiles measured along the broken red line in Figure 3 with box *C*. The intensities of each profile are normalized to the corresponding maximum.

**Figure 6 ijerph-18-01789-f006:**
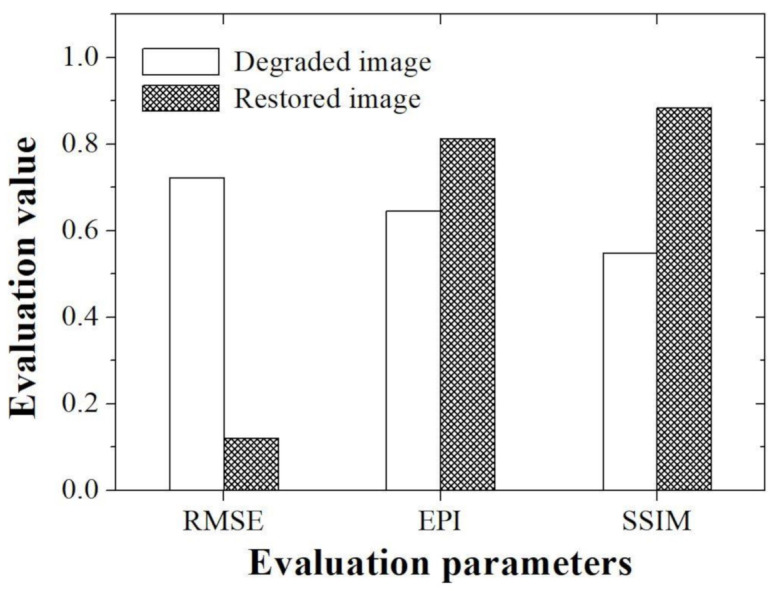
Graph of the evaluation parameters including the RMSE, EPI, and SSIM between the degraded and restored images.

**Figure 7 ijerph-18-01789-f007:**
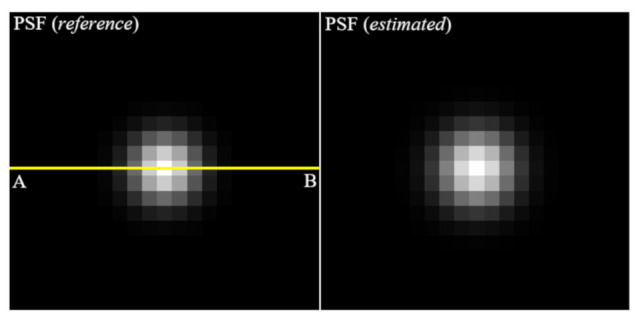
The images of PSF (reference), which is used by convolving the clean image to make the blurred image and PSF (estimated), which was estimated using the proposed algorithm.

**Figure 8 ijerph-18-01789-f008:**
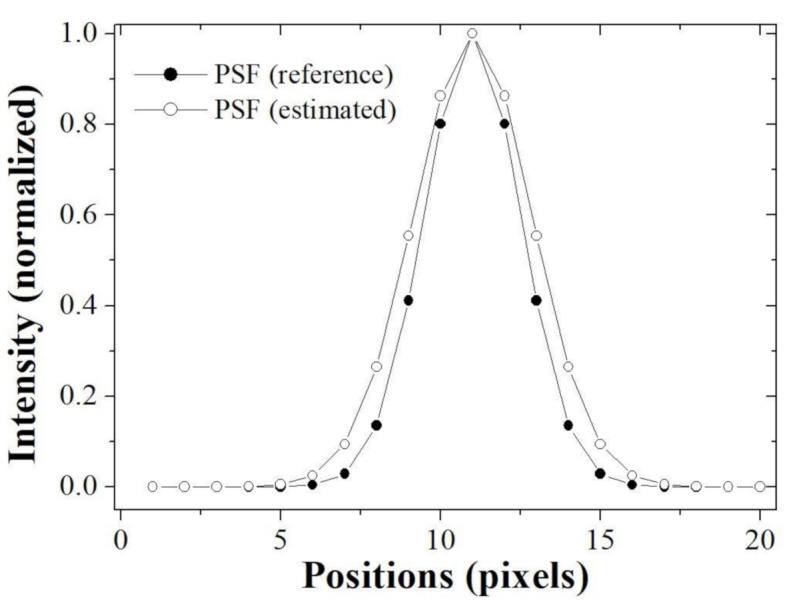
Profiles of the PSF (reference) and PSF (estimated) images measured along the line AB in the Figure 7.

**Figure 9 ijerph-18-01789-f009:**
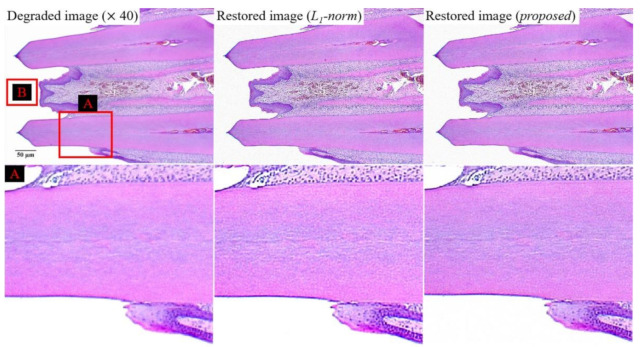
Experiment results for the degraded image (magnification of 40), restored image (*l*_1_-norm-based), and image restored using the proposed method.

**Figure 10 ijerph-18-01789-f010:**
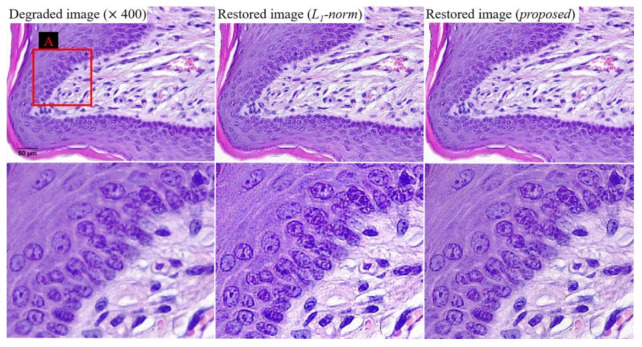
Examples of degraded image (magnification of 400), restored image (*l*_1_-norm-based), and image restored using the proposed method.

**Figure 11 ijerph-18-01789-f011:**
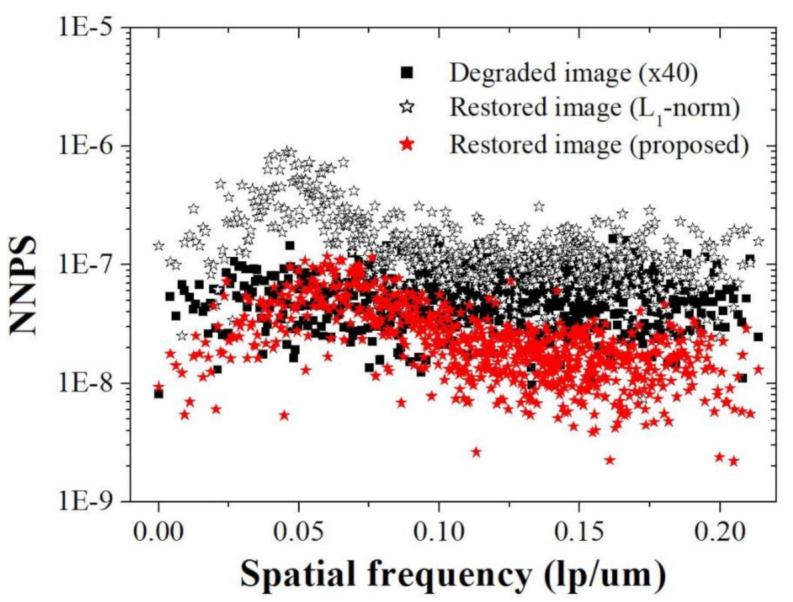
NNPS results (obtained using ROI B in Figure 9) of the degraded image (×40), restored image (*l*_1_-norm), and restored image (proposed) at all spatial frequencies (lp/μm).

**Figure 12 ijerph-18-01789-f012:**
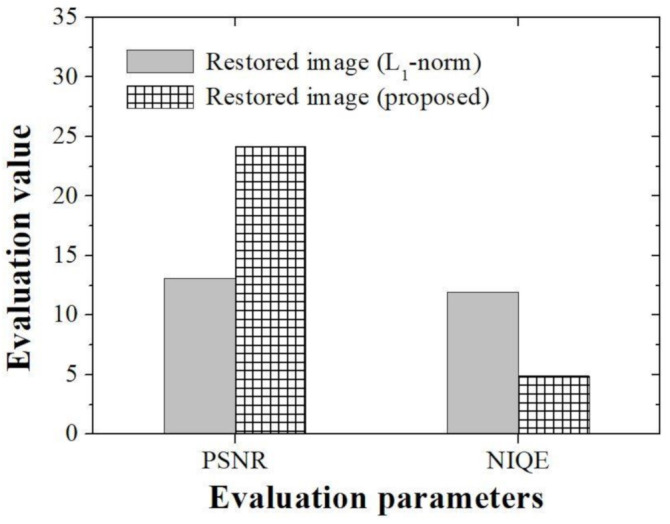
Graph of the evaluation parameters including the PSNR and NIQE using experimental image for *l*_1_-norm-based and proposed restoration methods.

## Data Availability

Not applicable.

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
