# Peer review of "Blind Deconvolution Based on Compressed Sensing with bi-l0-l2-norm Regularization in Light Microscopy Image"

_ijerph, 2021, doi:10.3390/ijerph18041789_

Round 1

Reviewer 1 Report

This manuscript studied the light microscopy image restoration in the framework of compressed sensing with bi-l0-l2regularization. The idea is interesting and the result is basically satisfactory. However, some other problems in the manuscript are still concerned in the following:

1. In the experiments, only the results of the proposed method were shown. This is not enough to validate its effectivity.Could the authors compare the proposed method with other state-of-the-art methods in the experiments?

2.There is a minor mistake in the equation of Figure 1.

3.How is the time cost of the proposed method?

4.This work is based on compressed sensing. In my opinion, more information on compressed sensing should be exposed in the text. Please refer to DOI: 10.1109/TIT.2006.871582, DOI: 10.1109/TSP.2007.900760, DOI: 10.1109/TGRS.2013.2245509...

5.2.2. Conditions for the simulation and experimentand 2.3. Evaluation factorsare suggested to be moved to 3. Results and Discussion.

6.The formats of the references are different in the manuscript, which should be the same.

Author Response

Dear reviewer,

Thank you for your useful comments and suggestions concerning our paper.

We attached response and revised manuscript files.

Best regards,

Ji-Youn Kim

Reviewer 2 Report

The authors described Blind deconvolution based on compressed sensing with bi-l0-l2-2 norm regularisation in light microscopy images. This is an interesting study in an area that needs investigating. The paper suffers, however, because some information of data is lacking. Minor revisions are needed.

Comments:

1. ⨂⨂ represents the two-dimensional (2D) convolution operator. (Line 41, 45 page 1)

 â¨‚⨂ means that two times convolution, or 2-dimensional blind deconvolution? Please explain in more detail.

2. In Figure 4, the difference between “Degraded image” and “Restored image” is not clear. I think "Restored" image should reduce noise after blind deconvolution. Isn’t it?

3. Line 222. Miss-spelling “oof “.

4. In Figure 9 and 10, a scale bar is needed. The authors only showed magnification (X4 or magnification of 400) in the text. The magnification of insets of “A” (both FIg. 9 and 10) are too small to see the details of the deconvoluted images. I can not see established images of noise reduction.

This protocol can be used more higher magnification (x!00) of image or low power view of images? If so, please include the comments in the discussion or include more images in Figure 10.

Is blind deconvolution also applicable for the image of fluorescent microscopy? If possible, include a fluorescent microscopy deconvolution images using this protocol.

Author Response

(The authors gave the same response as above.)

Reviewer 3 Report

Thank you for sharing your work. My comments/questions are as follows:

  1. Please provide some numerical findings in the abstract section of the claimed improvement.
  2. In the abstract, it is claimed that, “.. the proposed method is expected to …. compensate the limitation of the imaging hardware in term of image quality.” Please explain how the method can overcome the limitation of the imaging hardware?
  3. Line 34: analyses should be analysis.
  4. Line 45 – 48: please add references.
  5. Line 68: What is meant by “unnatural image” and “natural image”(line 72)?
  6. Why authors have selected the b1-l0-l2 norm regularization ?
  7. How the value of the optimization parameters are obtained ? (line 112, alpha = 500, line 113 beta_f = 0.25 etc.). The regularization strength depends on those parameters and therefore the quality of the image including noise.
  8. Line 118: What is the value of the tolerance and how it is determined ?
  9. Section 2.3: the description of RMSE, EPI, SSIM are unnecessary. They are standard image quality parameters and appropriate references should be enough.
  10. Line 205, What is meant by “focal formation” ?
  11. Please add a color scale with all images.
  12. Why the simulation images were normalized ? Restored image after deconvolution operation often have different intensity distribution. MTF would be an ideal measure of spatial resolution and contrast. Profile plot with normalized intensity distribution doesn’t convey much information. Please consider to add MTF for simulation (and experimental image if possible).
  13. SSIM is usually appropriate for images with complex structure. Numerical STAR doesn’t have any complex structure.
  14. Line 235 – 246, 252 – 267, 274 – 279 are repetition of standard definition of RMSE, EPI etc. Please use this section to discuss your results only.
  15. 8: Please explain, why the estimated PSF have underestimated performances ?
  16. Fig 9 & Fig 10: L1 – norm is not much different as compared to proposed method. It would be interesting to see the comparison with L1 -norm method for other image quality (SSIM, RMSE etc.). In addition to L1, the proposed method needs to compare with other  deconvolution algorithms.

Author Response

(The authors gave the same response as above.)

Reviewer 4 Report

Dear authors:

This is a well-organized paper on an interesting and relevant topic. However, there are some issues that have to be addressed:

  • A comparative study (in terms of RMSE, EPI and SSIM improvement) with other state-of-the-art methods is required in order to validate the proposed technique. State the percentage improvement by proposed work, compared with existing works.
  • How many images were used in the experiments? 

  • Results should be presented for more than one image.

- Rewrite the sentence: page 2, lines 74-77

“ The objective of this study ...and experimental images”

- Rewrite page 5 line 181: “obtained from on image,”

Author Response

(The authors gave the same response as above.)

Round 2

Reviewer 1 Report

All my questions have been solved.

Reviewer 3 Report

Thank you, for responding to the comments/questions and update the manuscript accordingly. I don't have any further questions.

Reviewer 4 Report

Authors addressed my comments and this version is a good improvement of the initial version.